# Pestilent relationship between smoking and hypertension or pulse pressure among males over 15 years in India: NFHS-5 Survey

Dhruvendra Lal[1], Amrit Kaur Virk[1], Anu Bhardwaj[1], Kavisha Kapoor Lal[2], Jayanta Bora[3‡], Anuradha Nadda[1‡], Sonu Goel [4‡]*

1 Department of Community Medicine, Dr B R Ambedkar State Institute of Medical Sciences (AIMS), Mohali, Punjab, India, 2 Department of Periodontics, Himachal Dental College, Sundernagar, Himachal Pradesh, India, 3 Founding & Executive Director, VART Consulting Pvt. Ltd, Delhi-NCR, Mumbai, India, 4 School of Medicine and Health Research Institute, University of Limerick, Limerick, Ireland

☯ These authors contributed equally to this work.
‡ JB, AN and SG also contributed equally to this work.
* sonu.goel@ul.ie

**Data Availability Statement:** All NFHS 5 India files are available from the database: https://www. dhsprogram.com/.

**Funding:** The author(s) received no specific funding for this work.

## Abstract

### Objective

The Global Adult Tobacco Survey conducted in India has divulged that 28.6% of the populace aged 15 years and above partakes in tobacco consumption in various modalities. Despite the availability of numerous studies on the correlation between smoking and hypertension, the nexus between tobacco smoking and hypertension remains enigmatic. Smoking has predominantly been linked to blood pressure, with scant investigations exploring the plausible association that may subsist between smoking and pulse pressure.

### Methodology

This study is based on secondary data analysis from the fifth National Family Health Survey (NFHS-5). 17 Field Agencies gathered information from 636,699 households, 724,115 women, and 101,839 men. The data related to only men was included and analysed in this present study.

### Results

Male participants had a mean age of 32.2+1.2 years, an average waist circumference of 80.4+12.2 cm, and mean systolic and diastolic blood pressure of 123.4+13.8 mmHg and 80.5+10.2 mmHg. Daily smokers had a slightly higher likelihood of hypertension compared to non-smokers (OR = 1.2, p <0.001). Male quitters had significantly lower odds of hypertension (OR = 0.9, p <0.001). Quitters had reduced odds of narrow pulse pressure but increased odds of wide pulse pressure (OR = 0.81 and 1.14, respectively).

**Competing interests:** The authors have declared that no competing interests exist.

## Conclusion

The study found that regular smoking was associated with hypertension, while factors such as age, obesity, urban dwelling, wealth, and tribal residence were linked to increased blood pressure. Male quitters had a lower likelihood of hypertension, and middle-aged men and those with central obesity showed distinct associations with deranged pulse pressure.

## Introduction

The national-level GATS 2 (Global Adult Tobacco Survey) conducted in India unveiled that a substantial proportion of the population, (28.6% of individuals aged 15 years and above) indulged in the consumption of tobacco in some form. Furthermore, the survey findings underscored the prevalence of tobacco smoking, impacting 11% of the surveyed populace [1]. In North India, smoking prevalence among males was approximately 20.4%, accompanied by varying degrees of nicotine dependence ranging from moderate to high. It is worth noting that smoking has been identified as the primary preventable factor contributing to cardiovascular diseases (CVDs) [2, 3].

Hypertension, being the foremost modifiable factor, plays a significant role in avoidable mortality and morbidity in India. It is closely associated with an elevated risk of cardiovascular diseases, contributing to nearly 23% of the total fatalities within the country [4]. A cross-sectional survey data from the fourth round (2015–2016) of National Family Health Survey (NFHS) has documented prevalence of hypertension in India as 11.3% (13.8% in males and 10.9% in females) [5]. Wide pulse pressure has gained increasing recognition as a salient contributory factor to cardiovascular disease. It emerges from the intricate interplay between cardiac ejection (stroke volume) and the complex dynamics of the arterial circulation. Hence, increased rigidity in larger vessels, notably the aorta, precipitates a discernible augmentation in pulse pressure [6].Pulse pressure can be obtained by the difference between the systolic and diastolic blood pressure of an individual [7].

Despite smoking being widely acknowledged as a risk factor for hypertension, contradictory findings have emerged from certain studies regarding the association between hypertension and smoking. Interestingly, several research studies have reported lower blood pressure levels among active smokers than individuals who have quit smoking [8]. Current smokers, both male and female, had lower rates of blood pressure control, with only 9.1% of males and 25% of females achieving control. Male former smokers had a similar rate of blood pressure control (37.6%) to never-smokers, while female former smokers had a lower rate (23.8%), indicating lingering risk even after quitting smoking [8]. A study conducted in Singapore unveiled an intriguing inverse relationship between cigarette smoking in males and systolic blood pressure, culminating in reductions of 1.3 mmHg (light smokers), 3.8 mmHg (moderate smokers), and 4.6 mmHg (heavy smokers) in comparison to nonsmokers. However, no definitive correlation was discerned between smoking and diastolic blood pressure [9]. A separate study indicated a noteworthy trend wherein extended periods of smoking cessation were associated with an elevated risk of hypertension, mainly observed among subgroups who maintained their weight or experienced weight gain after quitting smoking. These findings suggest smoking cessation could lead to increased blood pressure, hypertension, or both [10]. The study findings elucidate that a six-week smoking cessation manifests in a notable reduction of awake systolic blood pressure (BP) and heart rate, plausibly attributable to the attenuation of sympathetic activity. Consequently, these results underscore the significance of smoking cessation in

mitigating cardiovascular risk, primarily by curtailing awake systolic BP and heart rate [11]. On the other hand, a few studies demonstrated that smoking acutely increases arterial stiffness and blood pressure in males with hypertension, and these effects persist for a longer duration in smokers without hypertension [10, 12].

The primary objective of this study is to evaluate the prevalence of hypertension and smoking among males aged 15 years and above in India. Additionally, it aims to investigate the intricate association of tobacco smoking with blood pressure and pulse pressure (both wide and narrow) while considering the impact of diverse social and demographic factors, apart from analysing the effect of quitting smoking on hypertension/pulse pressure.

## Methodology

### Study design

This study is based on secondary data analysis from the fifth National Family Health Survey (NFHS-5), the Indian version of Demographic and Health Survey carried out periodically in over 90 countries across the globe. The NFHS is a collaborative project of the International Institute for Population Sciences (IIPS), Mumbai, India; ICF, Calverton, Maryland, USA and the East-West Center, Honolulu, Hawaii, USA. The Ministry of Health and Family Welfare (MOHFW), Government of India has designated IIPS as the nodal agency and responsible for providing coordination and technical guidance for the NFHS. NFHS is funded by the United States Agency for International Development (USAID) with supplementary support from United Nations Children's Fund (UNICEF). NFHS uses four survey questionnaires—household, woman's, man's, and biomarker, in 19 languages using Computer Assisted Personal Interviewing (CAPI). NFHS-5 fieldwork for India was conducted in two phases. Phase-I was started on 17 June 2019 to 30 January 2020 and covered 17 states and 5 Union Territories. Phase-II was initiated on 2 January 2020 to 30 April 2021 and covered 11 states and 3 Union Territories. 17 Field Agencies gathered information from 636,699 households, 724,115 women, and 101,839 men.

### Data source

The present analysis is based on the men's questionnaire of NFHS-5, male dataset, where the effect of smoking was seen on blood pressure and pulse pressure (dependent variable) against various independent variables which could have affected the blood pressure values. Gender roles and societal norms surrounding smoking differ between males and females in India. Studying males can provide insights into the social and cultural influences that shape smoking behaviours more accurately, as their numbers may predict the actual burden. Historically, smoking has been more socially accepted and prevalent among men. At the same time, women have faced a more substantial social stigma associated with smoking, and their inclusion may produce a biased result. The total sample size for this analysis was 101839 (males over 15 years of age). However, due to missing values (which could not be included during data analysis since these values would have altered the results and interpretation), each variable in this analysis has a different sample size, mentioned in the result section.

### Ethical approval

This research utilized secondary data, excluding any direct involvement of patients and the general public. As this study relied on an anonymous publicly available dataset without any identifiable information concerning the survey participants, the requirement for an ethics statement was obviated. Nevertheless, ethical approval was obtained from the Institutional

Ethics Committee of the Post Graduate Institute of Medical Education and Research (PGI-MER), Chandigarh (IEC-08/2022-2535 dated 17.08.2022).

## Definitions

Based on the established guidelines set forth by the World Health Organization (WHO) and American Heart Association, the definition of raised blood pressure in this study entails systolic blood pressure equal to or exceeding 140 mm Hg and/or diastolic blood pressure equal to or surpassing 90 mm Hg. Furthermore, the male participants were meticulously stratified into distinct categories reflective of the various stages of hypertension, namely Normal (<120 mm Hg), Pre-hypertension (121 to 129 mm Hg), Stage 1 hypertension (130 to 139 mm Hg), Stage 2 hypertension (140 to 179 mm Hg), and hypertensive crisis (>180 mm Hg), as per the prevailing guidelines [13, 14].

The difference between the mean systolic and diastolic blood pressure values gave the Pulse Pressure (PP). These values were categorized as low or narrow (<40mmHg), normal (40-60mmHg) and high or wide (>60mmHg) [15–18].

## Statistical analysis

Hypertension and deranged pulse pressure were considered as the outcome variable of this study. Three blood pressure readings were taken in NFHS, but mean of only two variables (2nd and 3rd reading) was taken as per the guidelines.

All statistical analyses were done using IBM SPSS version 20 for windows (IBM, Armonk, Ney York, United States). The data has been presented as means ± standard deviations, frequencies, and percentages. Association was calculated using chi-square test, and t-test and binary logistic regression along with multinomial logistic regression were used for calculating the odds ratio. The data set was checked for good fit using the Hosmer and Lemeshow Test and Goodness of Fit test, and the overall prediction of the model was made using sensitivity and specificity. Chi quare test was used for assessing the association between hypertension and pulse pressure with age, place of residence (urban/rural), wealth index (5 categories), education level (4 categories), ethnicity (4 categories), central obesity, smoking status, history of quitting smoking and currently cigarette/bidi smoking (3 categories). Binary logistic regression was used to calculate odds of raised blood pressure with age range (3 categories), central obesity, education (4 categories), area of residence (urban/rural), wealth index (5 categories), ethnicity (4 categories), smoking status, frequency of smoking (3 categories) and history of quitting smoking. Similarly, multinomial logistic regression was used to estimate and assess the adjusted associations between narrow and wide pulse pressure the already described variables. The level of significance was set at a 95% confidence interval and p value of less than 0.05 was considered as significant.

## Results

A comprehensive cohort of 101,839 male participants took part in the NFHS-5 study. Due to the presence of missing values in the dataset, the sample size varied for each variable and excluded these missing values during the analysis, as described in the methodology.

Table 1 presents noteworthy findings, indicating that the average age of all male participants was 32.2+1.2 years. The mean waist circumference was measured at 80.4+12.2 cm, accompanied by mean systolic and diastolic blood pressure readings of 123.4+13.8 mmHg and 80.5+10.2 mmHg, respectively. Notably, the data from Table 1 reveals a significant difference in mean blood pressure (systolic and diastolic) between smokers and non-smokers, with smokers exhibiting higher values.

**Table 1. Means of various variables of the target population.**

| Variable | N | Mean | SD |
|---|---|---|---|
| Age | 101839 | 32.2 | 1.2 |
| Waist Circumference (cm) | 96018 | 80.4 | 12.2 |
| Hip Circumference (cm) | 96014 | 89.6 | 10.5 |
| Systolic BP (mmHg) | 88923 | 123.4 | 13.8 |
| Diastolic BP (mmHg) | 92805 | 80.5 | 10.2 |
| Pulse Pressure | 88807 | 42.9 | 9.5 |
| Systolic BP in Non-Smokers* | 69928 | 123.0 | 13.7 |
| Systolic BP in Smokers* | 18995 | 124.6 | 14.2 |
| Diastolic BP in Non-Smokers* | 72798 | 80.2 | 10.2 |
| Diastolic BP in Smokers* | 20007 | 81.7 | 10.0 |
| Pulse Pressure in Non-Smokers** | 69836 | 42.9 | 9.4 |
| Pulse Pressure in Smokers** | 18971 | 42.8 | 9.8 |

* p value <0.001

** p value = 0.248

Table 2 elucidates significant associations between age and the severity of hypertension and pulse pressure among males. Age exhibits a concordant relationship with hypertension, while a more pronounced link is observed between advanced age and wide pulse-pressure. Notably, males in the intermediate age range of 30 to 45 years manifest a heightened prevalence of narrow pulse pressure. Urban dwellers display a considerably elevated incidence of hypertension (Stage I and Stage II). The wealth index demonstrates a direct correlation with all stages of hypertension, evincing a greater prevalence among affluent males than those in economically disadvantaged strata. Narrow pulse pressure is notably more prevalent among males from lower socio-economic backgrounds, whereas wide pulse pressure predominates among their higher socio-economic counterparts. Education level exhibits significant associations with hypertension and pulse pressure, with males lacking formal education showing a higher prevalence. Tribal communities showed an elevated prevalence of hypertension, while individuals without a designated caste demonstrated a stronger association with wide than narrow pulse pressure. Central obesity exhibits a significant association with hypertension and pulse pressure, with all stages of hypertension showing greater prevalence among males with central obesity. Additionally, wide pulse pressure demonstrates a significant association with central obesity.

Furthermore, all forms of tobacco smoking demonstrate a significant association with hypertension and pulse pressure, with male smokers revealing a higher prevalence of hypertension, narrow pulse pressure, and wide pulse pressure. Stage I hypertension is more prevalent among individuals who have never attempted to cease smoking. Male quitters exhibit a higher prevalence of narrow pulse pressure, while non-quitters exhibit a higher prevalence of wide pulse pressure. Daily smokers show a significant association with hypertension.

Table 3 uncovers noteworthy insights, indicating that males aged over 45 years exhibit significantly higher odds (OR = 1.9, CI = 1.8 to 2.1) of having hypertension compared to those in the 15 to 30-year age group. The study highlights that males with central obesity are significantly more prone to hypertension than their non-obese counterparts (OR = 1.8, CI = 1.7 to 2.0). Rural males have lower odds of being diagnosed with hypertension than urban males. Furthermore, wealthier males face a significantly higher risk of hypertension than individuals with lower socio-economic status (OR = 1.3, CI = 1.2 to 1.5). The study demonstrates that

**Table 2. Distribution between various variables with stages of hypertension and pulse pressure among the target population.**

| | | | HTN Staging | | | | | Total | Chi Square Value | Pulse Pressure | | | Total | Chi Square Value |
|---|---|---|---|---|---|---|---|---|---|---|---|---|---|---|
| | | | Normal | Pre HTN | Stage 1 HTN | Stage 2 HTN | HTN Crisis | | P Value | Normal | Narrow | Wide | | P Value |
| **Age** | **15 to 30 years** | N | 20253 | 13714 | 5182 | 1714 | 28 | 40891 | 6631.1 | 25427 | 14085 | 1491 | 41003 | 736.6 |
| | | % | 49.5% | 33.5% | 12.7% | 4.2% | 0.1% | 100.0% | <0.001 | 62.0% | 34.4% | 3.6% | 100.0% | <0.001 |
| | **30 to 45 years** | N | 10201 | 10787 | 6375 | 3640 | 149 | 31152 | | 17801 | 12182 | 1332 | 31315 | |
| | | % | 32.7% | 34.6% | 20.5% | 11.7% | 0.5% | 100.0% | | 56.8% | 38.9% | 4.3% | 100.0% | |
| | **45 to 60 years** | N | 3720 | 3982 | 3056 | 2926 | 191 | 13875 | | 8002 | 4783 | 1175 | 13960 | |
| | | % | 26.8% | 28.7% | 22.0% | 21.1% | 1.4% | 100.0% | | 57.3% | 34.3% | 8.4% | 100.0% | |
| Total | | N | 34174 | 28483 | 14613 | 8280 | 368 | 85918 | | 51230 | 31050 | 3998 | 86278 | |
| | | % | 39.8% | 33.2% | 17.0% | 9.6% | 0.4% | 100.0% | | 59.4% | 36.0% | 4.6% | 100.0% | |
| **Type of place of residence** | **Urban** | N | 8083 | 7304 | 4010 | 2329 | 117 | 21843 | 128.5 | 12985 | 7911 | 1069 | 21965 | 3.8 |
| | | % | 37.0% | 33.4% | 18.4% | 10.7% | 0.5% | 100.0% | <0.001 | 59.1% | 36.0% | 4.9% | 100.0% | 0.15 |
| | **Rural** | N | 26091 | 21179 | 10603 | 5951 | 251 | 64075 | | 38245 | 23139 | 2929 | 64313 | |
| | | % | 40.7% | 33.1% | 16.5% | 9.3% | 0.4% | 100.0% | | 59.5% | 36.0% | 4.6% | 100.0% | |
| Total | | N | 34174 | 28483 | 14613 | 8280 | 368 | 85918 | | 51230 | 31050 | 3998 | 86278 | |
| | | % | 39.8% | 33.2% | 17.0% | 9.6% | 0.4% | 100.0% | | 59.4% | 36.0% | 4.6% | 100.0% | |
| **Wealth index combined** | **Poorest** | N | 7317 | 5552 | 2661 | 1402 | 58 | 16990 | 493.5 | 10253 | 6073 | 777 | 17103 | 64.5 |
| | | % | 43.1% | 32.7% | 15.7% | 8.3% | 0.3% | 100.0% | <0.001 | 59.9% | 35.5% | 4.5% | 100.0% | <0.001 |
| | **Poorer** | N | 8147 | 6227 | 3062 | 1665 | 87 | 19188 | | 11400 | 6972 | 843 | 19215 | |
| | | % | 42.5% | 32.5% | 16.0% | 8.7% | 0.5% | 100.0% | | 59.3% | 36.3% | 4.4% | 100.0% | |
| | **Middle** | N | 7459 | 6084 | 3069 | 1703 | 76 | 18391 | | 10692 | 6914 | 812 | 18418 | |
| | | % | 40.6% | 33.1% | 16.7% | 9.3% | 0.4% | 100.0% | | 58.1% | 37.5% | 4.4% | 100.0% | |
| | **Richer** | N | 6364 | 5674 | 2956 | 1796 | 77 | 16867 | | 10006 | 6171 | 801 | 16978 | |
| | | % | 37.7% | 33.6% | 17.5% | 10.6% | 0.5% | 100.0% | | 58.9% | 36.3% | 4.7% | 100.0% | |
| | **Richest** | N | 4887 | 4946 | 2865 | 1714 | 70 | 14482 | | 8879 | 4920 | 765 | 14564 | |
| | | % | 33.7% | 34.2% | 19.8% | 11.8% | 0.5% | 100.0% | | 61.0% | 33.8% | 5.3% | 100.0% | |
| Total | | N | 34174 | 28483 | 14613 | 8280 | 368 | 85918 | | 51230 | 31050 | 3998 | 86278 | |
| | | % | 39.8% | 33.2% | 17.0% | 9.6% | 0.4% | 100.0% | | 59.4% | 36.0% | 4.6% | 100.0% | |
| **Educational level** | **No education** | N | 3819 | 3392 | 1940 | 1217 | 63 | 10431 | 318.4 | 6053 | 3850 | 600 | 10503 | 69.4 |
| | | % | 36.6% | 32.5% | 18.6% | 11.7% | 0.6% | 100.0% | <0.001 | 57.6% | 36.7% | 5.7% | 100.0% | <0.001 |
| | **Primary** | N | 3768 | 3321 | 1818 | 1154 | 60 | 10121 | | 5833 | 3803 | 508 | 10144 | |
| | | % | 37.2% | 32.8% | 18.0% | 11.4% | 0.6% | 100.0% | | 57.5% | 37.5% | 5.0% | 100.0% | |
| | **Secondary** | N | 21143 | 16497 | 8263 | 4552 | 193 | 50648 | | 30306 | 18198 | 2275 | 50779 | |
| | | % | 41.7% | 32.6% | 16.3% | 9.0% | 0.4% | 100.0% | | 59.7% | 35.8% | 4.5% | 100.0% | |
| | **Higher** | N | 5444 | 5273 | 2592 | 1357 | 52 | 14718 | | 9038 | 5199 | 615 | 14852 | |
| | | % | 37.0% | 35.8% | 17.6% | 9.2% | 0.4% | 100.0% | | 60.9% | 35.0% | 4.1% | 100.0% | |
| Total | | N | 34174 | 28483 | 14613 | 8280 | 368 | 85918 | | 51230 | 31050 | 3998 | 86278 | |
| | | % | 39.8% | 33.2% | 17.0% | 9.6% | 0.4% | 100.0% | | 59.4% | 36.0% | 4.6% | 100.0% | |
| **Ethnicity** | **Caste** | N | 28199 | 23089 | 11747 | 6611 | 288 | 69934 | 108 | 41389 | 25689 | 3184 | 70262 | 64.6 |
| | | % | 40.3% | 33.0% | 16.8% | 9.5% | 0.4% | 100.0% | <0.001 | 58.9% | 36.6% | 4.5% | 100.0% | <0.001 |
| | **Tribe** | N | 4389 | 4167 | 2271 | 1315 | 60 | 12202 | | 7556 | 4062 | 598 | 12216 | |
| | | % | 36.0% | 34.2% | 18.6% | 10.8% | 0.5% | 100.0% | | 61.9% | 33.3% | 4.9% | 100.0% | |
| | **No caste / tribe** | N | 1396 | 1068 | 509 | 309 | 19 | 3301 | | 1994 | 1140 | 184 | 3318 | |
| | | % | 42.3% | 32.4% | 15.4% | 9.4% | 0.6% | 100.0% | | 60.1% | 34.4% | 5.5% | 100.0% | |
| | **Don't know** | N | 190 | 159 | 86 | 45 | 1 | 481 | | 291 | 159 | 32 | 482 | |
| | | % | 39.5% | 33.1% | 17.9% | 9.4% | 0.2% | 100.0% | | 60.4% | 33.0% | 6.6% | 100.0% | |
| Total | | N | 34174 | 28483 | 14613 | 8280 | 368 | 85918 | | 51230 | 31050 | 3998 | 86278 | |
| | | % | 39.8% | 33.2% | 17.0% | 9.6% | 0.4% | 100.0% | | 59.4% | 36.0% | 4.6% | 100.0% | |

**Table 2.** (Continued)

| | | | Normal | Pre HTN | Stage 1 HTN | Stage 2 HTN | HTN Crisis | Total | P Value | Normal | Narrow | Wide | Total | P Value |
|---|---|---|---|---|---|---|---|---|---|---|---|---|---|---|
| | | | | | HTN Staging | | | Total | Chi Square Value | | Pulse Pressure | | Total | Chi Square Value |
| **Central Obesity** | **Normal** | N | 30067 | 23088 | 10721 | 5250 | 204 | 69330 | 3347.6 | 41381 | 25126 | 3002 | 69509 | 84.4 |
| | | % | 43.4% | 33.3% | 15.5% | 7.6% | 0.3% | 100.0% | <0.001 | 59.5% | 36.1% | 4.3% | 100.0% | <0.001 |
| | **Present** | N | 3553 | 4694 | 3426 | 2768 | 151 | 14592 | | 8686 | 5165 | 893 | 14744 | |
| | | % | 24.3% | 32.2% | 23.5% | 19.0% | 1.0% | 100.0% | | 58.9% | 35.0% | 6.1% | 100.0% | |
| Total | | N | 33620 | 27782 | 14147 | 8018 | 355 | 83922 | | 50067 | 30291 | 3895 | 84253 | |
| | | % | 40.1% | 33.1% | 16.9% | 9.6% | 0.4% | 100.0% | | 59.4% | 36.0% | 4.6% | 100.0% | |
| **Smoking** | **Non Smoker** | N | 27616 | 22216 | 11222 | 6291 | 263 | 67608 | 178.8 | 40469 | 24316 | 3085 | 67870 | 11 |
| | | % | 40.8% | 32.9% | 16.6% | 9.3% | 0.4% | 100.0% | <0.001 | 59.6% | 35.8% | 4.5% | 100.0% | 0.004 |
| | **Smoker** | N | 6558 | 6267 | 3391 | 1989 | 105 | 18310 | | 10761 | 6734 | 913 | 18408 | |
| | | % | 35.8% | 34.2% | 18.5% | 10.9% | 0.6% | 100.0% | | 58.5% | 36.6% | 5.0% | 100.0% | |
| Total | | N | 34174 | 28483 | 14613 | 8280 | 368 | 85918 | | 51230 | 31050 | 3998 | 86278 | |
| | | % | 39.8% | 33.2% | 17.0% | 9.6% | 0.4% | 100.0% | | 59.4% | 36.0% | 4.6% | 100.0% | |
| **History of quitting smoking in the past 01 year** | **No** | N | 9846 | 9222 | 4865 | 2880 | 148 | 26961 | 38.1 | 16023 | 9756 | 1350 | 27129 | 91.5 |
| | | % | 36.5% | 34.2% | 18.0% | 10.7% | 0.5% | 100.0% | <0.001 | 59.1% | 36.0% | 5.0% | 100.0% | <0.001 |
| | **Yes** | N | 4316 | 3474 | 1856 | 1145 | 50 | 10841 | | 5997 | 4487 | 443 | 10927 | |
| | | % | 39.8% | 32.0% | 17.1% | 10.6% | 0.5% | 100.0% | | 54.9% | 41.1% | 4.1% | 100.0% | |
| Total | | N | 14162 | 12696 | 6721 | 4025 | 198 | 37802 | | 22020 | 14243 | 1793 | 38056 | |
| | | % | 37.5% | 33.6% | 17.8% | 10.6% | 0.5% | 100.0% | | 57.9% | 37.4% | 4.7% | 100.0% | |
| **Currently smokes cigarettes** | **Not at all** | N | 29907 | 24212 | 12391 | 7005 | 298 | 73813 | 153.4 | 44014 | 26671 | 3418 | 74103 | 1.5 |
| | | % | 40.5% | 32.8% | 16.8% | 9.5% | 0.4% | 100.0% | <0.001 | 59.4% | 36.0% | 4.6% | 100.0% | 0.822 |
| | **Every day** | N | 1884 | 2064 | 1073 | 643 | 26 | 5690 | | 3415 | 2037 | 279 | 5731 | |
| | | % | 33.1% | 36.3% | 18.9% | 11.3% | 0.5% | 100.0% | | 59.6% | 35.5% | 4.9% | 100.0% | |
| | **Some days** | N | 2383 | 2207 | 1149 | 632 | 44 | 6415 | | 3801 | 2342 | 301 | 6444 | |
| | | % | 37.1% | 34.4% | 17.9% | 9.9% | 0.7% | 100.0% | | 59.0% | 36.3% | 4.7% | 100.0% | |
| Total | | N | 34174 | 28483 | 14613 | 8280 | 368 | 85918 | | 51230 | 31050 | 3998 | 86278 | |
| | | % | 39.8% | 33.2% | 17.0% | 9.6% | 0.4% | 100.0% | | 59.4% | 36.0% | 4.6% | 100.0% | |
| **Do you currently smoke bidis** | **Not at all** | N | 30570 | 25210 | 12754 | 7127 | 304 | 75965 | 122.7 | 45496 | 27300 | 3461 | 76257 | 34.4 |
| | | % | 40.2% | 33.2% | 16.8% | 9.4% | 0.4% | 100.0% | <0.001 | 59.7% | 35.8% | 4.5% | 100.0% | <0.001 |
| | **Every day** | N | 2580 | 2268 | 1365 | 833 | 46 | 7092 | | 4048 | 2731 | 376 | 7155 | |
| | | % | 36.4% | 32.0% | 19.2% | 11.7% | 0.6% | 100.0% | | 56.6% | 38.2% | 5.3% | 100.0% | |
| | **Some days** | N | 1024 | 1005 | 494 | 320 | 18 | 2861 | | 1686 | 1019 | 161 | 2866 | |
| | | % | 35.8% | 35.1% | 17.3% | 11.2% | 0.6% | 100.0% | | 58.8% | 35.6% | 5.6% | 100.0% | |
| Total | | N | 34174 | 28483 | 14613 | 8280 | 368 | 85918 | | 51230 | 31050 | 3998 | 86278 | |
| | | % | 39.8% | 33.2% | 17.0% | 9.6% | 0.4% | 100.0% | | 59.4% | 36.0% | 4.6% | 100.0% | |

smokers' odds of being diagnosed with hypertension are nearly equivalent to non-smokers' (OR = 1.0, CI = 0.9 to 1.1). However, individuals who smoke cigarettes daily have significantly higher chances of being diagnosed with hypertension than non-smokers (OR = 1.17, CI = 1.1 to 1.3). Additionally, the analysis reveals that male individuals who quit smoking have significantly lower odds of being diagnosed with hypertension than those who continue smoking (OR = 0.86, CI = 0.8 to 0.9).

The right side of Table 3 presents findings regarding pulse pressure. Narrow pulse pressure is more commonly observed among 30 to 45-year-old males (OR = 1.1, CI = 1.1 to 1.2) than males with normal pulse pressure. Conversely, wide pulse pressure is less prevalent in the

Table 3. Relationship between blood pressure/pulse pressure and smoking and other factors among males above 15 years of age in India.

| | Raised Blood Pressure* | | | | Narrow Pulse Pressure ** | | | | Wide Pulse Pressure ** | | | |
|---|---|---|---|---|---|---|---|---|---|---|---|---|
| | P value | Odds Ratio | 95% CI | | P value | Odds Ratio | 95% CI | | P value | Odds Ratio | 95% CI | |
| | | | Lower Bound | Upper Bound | | | Lower Bound | Upper Bound | | | Lower Bound | Upper Bound |
| **Age** | | | | | | | | | | | | |
| 15 to 30 years | | 1 | | | <0.001 | 0.85 | 0.8 | 0.91 | <0.001 | 0.43 | 0.37 | 0.49 |
| 30 to 45 years | <0.001 | 1.51 | 1.43 | 1.58 | <0.001 | 1.14 | 1.08 | 1.21 | <0.001 | 0.52 | 0.47 | 0.59 |
| > 45 years | <0.001 | 1.94 | 1.82 | 2.07 | | 1 | | | | 1 | | |
| **Central Obesity Absent** | | 1 | | | <0.001 | 1.14 | 1.07 | 1.21 | 0.03 | 0.87 | 0.76 | 0.99 |
| **Central Obesity Present** | <0.001 | 1.84 | 1.72 | 1.97 | | 1 | | | | 1 | | |
| **Educational level** | | | | | | | | | | | | |
| No education | | 1 | | | 0.31 | 1.05 | 0.96 | 1.16 | 0.36 | 1.1 | 0.89 | 1.37 |
| Primary | 0.64 | 1.02 | 0.95 | 1.09 | 0.13 | 1.08 | 0.98 | 1.18 | 0.95 | 1.01 | 0.81 | 1.25 |
| Secondary | 0.96 | 1 | 0.94 | 1.06 | 0.24 | 1.05 | 0.97 | 1.13 | 0.97 | 1 | 0.83 | 1.19 |
| Higher | 0.34 | 1.05 | 0.95 | 1.15 | | 1 | | | | 1 | | |
| **Area** | | | | | | | | | | | | |
| Urban | | 1 | | | 0.09 | 1.05 | 0.99 | 1.12 | 0.92 | 1.01 | 0.88 | 1.15 |
| Rural | 0.03 | 0.94 | 0.88 | 0.99 | | 1 | | | | 1 | | |
| **Wealth index** | | | | | | | | | | | | |
| Poorest | | 1 | | | <0.001 | 1.15 | 1.04 | 1.26 | 0.7 | 0.96 | 0.78 | 1.19 |
| Poorer | 0.5 | 0.98 | 0.92 | 1.04 | <0.001 | 1.18 | 1.08 | 1.29 | 0.71 | 0.96 | 0.79 | 1.18 |
| Middle | 0.04 | 1.07 | 1 | 1.15 | <0.001 | 1.25 | 1.14 | 1.36 | 0.72 | 0.96 | 0.79 | 1.18 |
| Richer | <0.001 | 1.13 | 1.05 | 1.22 | <0.001 | 1.17 | 1.07 | 1.28 | 0.78 | 1.03 | 0.84 | 1.25 |
| Richest | <0.001 | 1.34 | 1.21 | 1.47 | | 1 | | | | 1 | | |
| **Ethnicity** | | | | | | | | | | | | |
| Caste | | 1 | | | 0.12 | 1.24 | 0.94 | 1.64 | 0.9 | 1.04 | 0.56 | 1.93 |
| Tribe | <0.001 | 1.32 | 1.24 | 1.39 | 0.78 | 1.04 | 0.79 | 1.38 | 0.72 | 1.12 | 0.6 | 2.11 |
| No Caste/tribe | 0.45 | 0.96 | 0.86 | 1.07 | 0.75 | 1.05 | 0.78 | 1.41 | 0.27 | 1.44 | 0.75 | 2.77 |
| Don't know | 0.91 | 0.98 | 0.75 | 1.29 | | 1 | | | | 1 | | |
| **Smoking status** | | | | | | | | | | | | |
| Non Smoker | | 1 | | | 0.16 | 1.08 | 0.97 | 1.2 | 0.38 | 1.11 | 0.88 | 1.4 |
| Smoker | 0.9 | 1.01 | 0.91 | 1.12 | | 1 | | | | 1 | | |
| **Currently smokes cigarettes** | | | | | | | | | | | | |
| Not at all | | 1 | | | 0.26 | 0.94 | 0.85 | 1.04 | 0.33 | 0.9 | 0.72 | 1.12 |
| Everyday | <0.001 | 1.17 | 1.06 | 1.29 | 0.13 | 0.94 | 0.87 | 1.02 | 0.69 | 0.96 | 0.81 | 1.15 |
| Some days | 0.13 | 1.08 | 0.98 | 1.2 | | 1 | | | | 1 | | |
| **Do you currently smoke bidis** | | | | | | | | | | | | |
| Not al all | | 1 | | | 0.66 | 1.02 | 0.93 | 1.13 | 0.04 | 0.8 | 0.64 | 0.99 |
| Everyday | 0.24 | 0.94 | 0.86 | 1.04 | 0.08 | 1.09 | 0.99 | 1.2 | 0.17 | 0.87 | 0.7 | 1.06 |
| Some days | 0.46 | 1.04 | 0.94 | 1.15 | | 1 | | | | 1 | | |
| **History of quitting smoking** | | | | | | | | | | | | |
| Did not tried quitting smoking | | 1 | | | <0.001 | 0.81 | 0.78 | 0.85 | 0.02 | 1.14 | 1.02 | 1.28 |
| Tried to quit smoking | <0.001 | 0.86 | 0.82 | 0.9 | | 1 | | | | 1 | | |

*Binary Logistic Regression          ** Multinomial Logistic Regression

(Continued)

**Table 3.** (Continued)

| | Raised Blood Pressure* | | | | Narrow Pulse Pressure ** | | | | Wide Pulse Pressure ** | | | |
|---|---|---|---|---|---|---|---|---|---|---|---|---|
| | P value | Odds Ratio | 95% CI | | P value | Odds Ratio | 95% CI | | P value | Odds Ratio | 95% CI | |
| | | | Lower Bound | Upper Bound | | | Lower Bound | Upper Bound | | | Lower Bound | Upper Bound |
| Reference category: Normal Blood Pressure | | | | | Reference category: Normal pulse Pressure | | | | | | | |
| Hosmer and Lemeshow Test: P value = 0.07 (Non Significant, hence indicating a good fit) | | | | | Goodness of Fit Test: p value = 0.5 (Non Significant, hence indicating a good fit) | | | | | | | |
| Overall Prediction of model is 62.4% with Sensitivity of 91.4% and Specificity of 14.3% | | | | | Model Fitting Information table: p value < 0.001 | | | | | | | |

younger age group of 15 to 30 years (OR = 0.4, CI = 0.4 to 0.5). Non-obese males have a significantly higher likelihood of narrow pulse pressure (OR = 1.1, CI = 1.1 to 1.2), while they are less prone to wide pulse pressure (OR = 0.9, CI = 0.7 to 0.9). The odds of narrow pulse pressure were significantly higher across all wealth index categories than males with normal pulse pressure. Furthermore, individuals who quit smoking have significantly lower odds of narrow pulse pressure and higher odds of wide pulse pressure (OR = 0.81, CI = 0.7 to 0.8 and OR = 1.14, CI = 1.0 to 1.2, respectively).

The study findings highlight Uttar Pradesh in India as having the highest prevalence of smokers (10.9%) and individuals with hypertension (12.1%). Moreover, Uttar Pradesh carries the highest burden of hypertensive smokers in the country (11.3%), followed by Rajasthan and Arunachal Pradesh (Fig 1). Fig 2 demonstrates a significant difference in blood pressure levels between smokers and non-smokers, indicating an association between smoking and altered blood pressure.

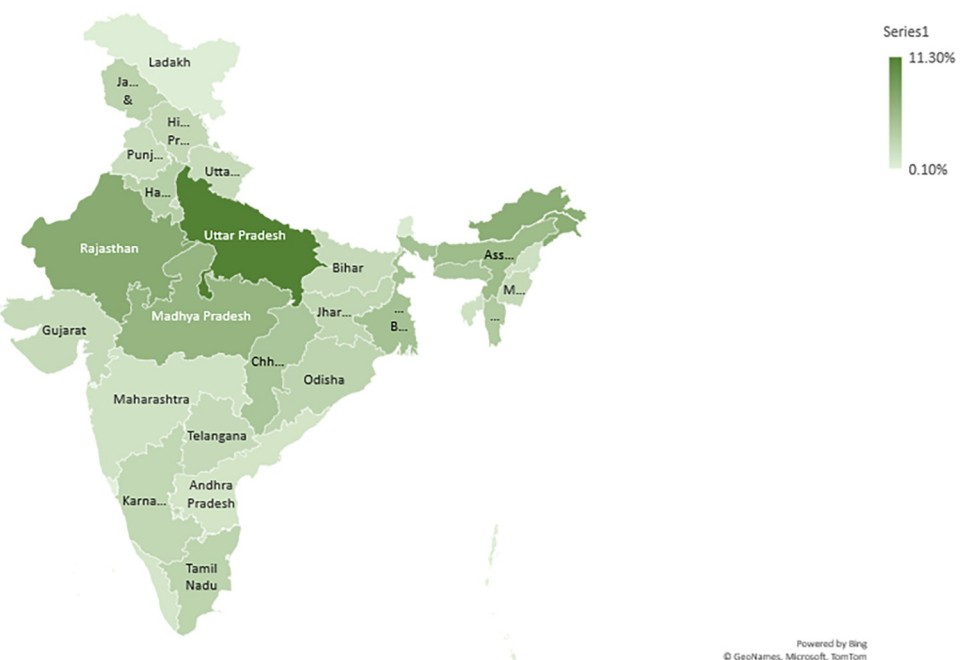

**Fig 1. State wise distribution of burden of increased blood pressure among male smokers >15 years in India.**

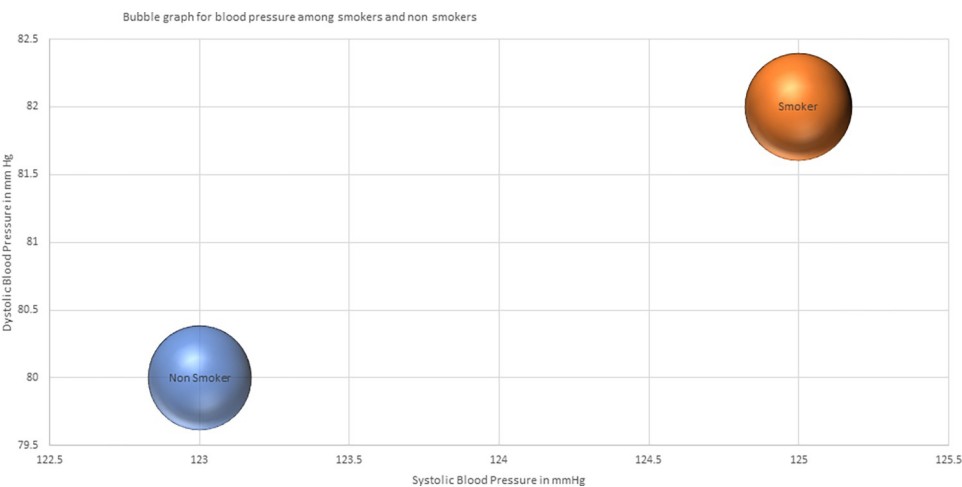

**Fig 2. Bubble graph with mean systolic, diastolic blood pressure among male smokers and non-smokers.**

## Discussion

The present study makes an attempt to find out the association of smoking and hypertension/ pulse pressure using data from the fifth round of NFHS (2019–2021).

Even though some research has been carried out earlier [9–12] to find out the association of smoking and hypertension, their relationship needs to be explored in detail. The association between smoking and pulse pressure however is very limited. Despite exhaustive literature search, the authors were not able to find any such study in India.

The present study elucidates that the overall odds of hypertension among present and past smokers evinced scarcely discernible disparity when compared to non-smokers (OR = 1.01, p = 0.90). These findings are consonant with the antecedent investigation by Ghosh et al in a secondary data analysis of NFHS-4 [5]. In contrast Singh et al reported from NFHS-4 that odds of male smokers having hypertension was 0.85 as compared to non-smokers [18].

Our study found that current smokers who indulge in daily cigarette consumption exhibit an increased likelihood of hypertension diagnosis when contrasted with a typical male (OR = 1.17). Conversely, a Chinese study among 28,577 men showed a diminished odds ratio of 0.88 among current smokers aged < 35 years. Whilst, men > 35 years displayed an odds ratio of 0.93 concerning the development of hypertension [19]. These intriguing disparities in outcomes underscore the importance of considering diverse demographic and geographical factors when investigating the relationship between smoking and hypertension. Indeed, the observed differences in the association between smoking and hypertension in the two studies could be attributed to various factors, including higher engagement in physical activities among Chinese individuals compared to Indians, variations in dietary habits between the two cultures, and the genetic predisposition of Indians towards hypertension.

Our results found that male individuals who quit smoking had a significantly lower likelihood of being diagnosed with hypertension than those who continued smoking (OR = 0.86). This suggests that quitting smoking may have a beneficial effect on reducing hypertension risk in this male population. This finding aligns with a study conducted by Lee et al, where they investigated the adjusted relative risk of hypertension among individuals who had quit smoking for different durations. According to their report, the relative risk of hypertension was 0.6 (95% CI 0.2 to 1.9) for those who had quit for less than one year, 1.5 (95% CI 0.8 to 2.8) for those who had quit for 1 to 3 years, and 3.5 (95% CI 1.7 to 7.4) for those who had quit for three

or more years, in comparison with current smokers [11]. These findings suggest that the risk of hypertension gradually decreases with increasing duration of smoking cessation. Even a relatively short period of smoking cessation could be associated with a reduced risk of hypertension, and the beneficial effects continue to increase with long-term abstinence from smoking. The implications of these findings should be taken into account for policy decisions on promoting smoking cessation programs to improve cardiovascular health and reduce the population's hypertension burden. Our study further demonstrated a significant association between increasing age and hypertension. It was seen that men aged > 45 years exhibited 1.94 times higher odds of having hypertension than males in the 15 to 30 years age group. These results are indicative of age being a critical factor contributing to the risk of hypertension in males.

Ghosh et al. reported the risk of hypertension as 6.7 times higher in the 45–49 age group than in the 15–19 age group [5]. It is worth noting that similar findings regarding the relationship between age and hypertension have been reported in other independent studies. The consistent presence of this association across different research reinforces the importance of age as a vital risk factor for hypertension. It highlights the need for effective age-specific interventions to address this public health concern [19].

Our study's results indicated that individuals from the wealthiest quintile had a 1.34 times higher likelihood of having hypertension than those in the poorest quintile. These findings are consistent with a study conducted by Geldsetzer et al., where they reported even higher odds of hypertension, namely 4.15 in the urban population and 3.01 in the rural population, using pooled data from District Level Household Survey-4 (DLHS-4) and Annual Health Survey. The increased risk of hypertension among the affluent population is attributed to lifestyle-related factors including improper dietary habits, lack of physical activities, and the urban way of life generally associated with high socio-economic status [20]. Additionally, factors like air pollution, psychosocial stress, and job-related stress in urban areas can adversely impact cardiovascular health.

The current study found that males with central obesity had 1.84 times higher odds of having hypertension compared to those without central obesity. This finding aligns with another study from Indonesia, which reported similar results with an odds ratio of 1.50 and a 95% confidence interval of 1.46–1.53 [21]. The relationship between hypertension and obesity is multifaceted and involves various complex mechanisms. The pathophysiology underlying hypertension due to obesity involves several interconnected factors, where the sympathetic nervous system becomes excessively active, leading to increased heart rate and blood vessel constriction, ultimately elevating blood pressure [22].

Our results for pulse pressure divulged a higher likelihood of narrow pulse pressure in middle-aged males, specifically those aged 30 to 45 years (OR = 1.14, p<0.001). Conversely, wide pulse pressure was less prevalent in the younger age group, i.e. 15 to 30 years old (OR = 0.43, p<0.001). Notably, a previous study from France corroborated these findings, reporting that in both sexes, the mean clinical pulse pressure widened as age advanced but intriguingly reached a noticeable plateau between 16 and 50 years of age. These observations underscore the significance of age-related variations in pulse pressure dynamics and warrant further exploration to elucidate the underlying physiological mechanisms contributing to these trends [23].

Wide pulse pressure is associated with deteriorating cardiac condition, which is less common among young adults and is physiologically seen among people with advanced age, pregnancy and well-conditioned athletes [24, 25]. The phenomenon of wide pulse pressure is also observed in some medical disorders, including aortic regurgitation, aortic sclerosis (both valvular pathologies), severe iron deficiency anaemia (causing diminished blood viscosity), arteriosclerosis (resulting in less elastic arteries), and hyperthyroidism (giving rise to heightened systolic pressure). This condition manifests as an increased difference between the systolic and

diastolic blood pressures, reflecting the underlying pathological alterations specific to these diverse clinical scenarios. The aetiology of this widened pulse pressure encompasses intricate hemodynamic perturbations involving alterations in cardiac valve function, blood viscosity, arterial compliance, and endocrine influences, collectively influencing the dynamics of blood flow and pressure within the cardiovascular system [7, 22, 26].

Narrow pulse pressure was significantly more common among non-obese males (OR = 1.14, p<0.001). A study by G De Pergolia et al. revealed that 24-hour Pulse Pressure values were significantly and positively associated with Body Mass Index, waist circumference and insulin levels among the participants [24].

In this study, quitters were seen to have less chances of a narrow pulse pressure and higher chances of wide pulse pressure with an odds ratio of 0.81 and 1.14 respectively. Narrow pulse pressures are a manifestation commonly encountered in diverse pathological conditions, each distinguished by specific pathophysiological perturbations. These maladies encompass heart failure, wherein the heart's contractile ability is compromised, leading to diminished pumping efficacy and subsequent reduction in stroke volume. Additionally, blood loss resulting from haemorrhage or traumatic events contributes to decreased blood volume, culminating in reduced cardiac output and stroke volume, thereby engendering a constricted pulse pressure. Furthermore, aortic stenosis, characterized by the narrowing of the aortic valve orifice, impedes blood flow from the left ventricle into the aorta during systole, resulting in diminished systolic pressure and consequently, a narrow pulse pressure. Lastly, the condition of cardiac tamponade arises due to the accumulation of fluid in the pericardial sac around the heart, exerting external pressure on the cardiac chambers and restricting diastolic filling. This impediment in filling causes a decline in stroke volume and systolic pressure, further contributing to a narrow pulse pressure [7, 22, 24].

Interestingly, our findings point a finger towards the state of Uttar Pradesh, India, emerging as a significant hotspot with the highest prevalence of both smokers (10.9%) and individuals diagnosed with hypertension (12.1%). Notably, Uttar Pradesh bears the heaviest burden of hypertensive smokers in the country, with a staggering 11.3% of the population affected, surpassing other states like Rajasthan and Arunachal Pradesh in this regard. These findings highlight the urgent need for targeted public health interventions in Uttar Pradesh to address the dual challenge of high smoking rates and hypertension prevalence. The analysis of NFHS-4 conducted by Gosh et al found that age-adjusted prevalence of hypertension in India was reported to be 11.3% (95% CI 11.16% to 11.43%) among individuals aged between 15 and 49 years. However, the proportion of the population affected by hypertension exhibited significant variation across different states (8.2% in Kerala to 20.3% in Sikkim) [5].

## Strengths

This broad and representative sample allows for a robust and comprehensive scrutiny of the impact of smoking on pulse pressure on a sizable sample size of males in India. This distinct approach of establishing a relationship between smoking and disrupted (wide or narrow) pulse pressure contributes valuable insights to the limited research addressing this specific aspect of male health in India and also provides a widened view of the effects of smoking on the cardiovascular system, which may require planning of longitudinal studies in this population group.

## Limitations

The study does confront a few inherent limitations, which includes the study's reliance on cross-sectional data, as furnished by the NFHS, which curtails the ability to establish causal

relationships between smoking and pulse pressure. Furthermore, the analysis was confined to solely the variables reported in the NFHS-5 dataset, inadvertently omitting some factors like behavioural and diet-related aspects.

## Conclusions

The research outcomes substantiate a robust linkage between smoking and hypertension, using the regression model and a significant difference was observed between the values of blood pressure among smokers and non-smokers using T-test. Furthermore, several factors like getting older, being centrally obese, living in urban areas, having a higher income, and belonging to a certain tribal community were significantly associated with higher blood pressure among men. Additionally, the study unveiled an intriguing connection of pulse pressure dynamics, wherein middle-aged men were significantly associated with the pertinently narrow pulse pressure, while males grappling with central obesity were more prone to have wide pulse pressure. Men who quit smoking showed a significant decrease in their likelihood of developing hypertension compared to those who continued smoking. Additionally, individuals who successfully quit smoking had a significantly lower chance of having narrow pulse pressure and a notably higher chance of having wide pulse pressure.

## Supporting information

**S1 Checklist. STROBE statement—Checklist of items that should be included in reports of *cross-sectional studies*.**
(DOCX)

**S1 Questionnaire.**
(DOCX)

## Author Contributions

**Conceptualization:** Dhruvendra Lal, Amrit Kaur Virk, Anu Bhardwaj, Kavisha Kapoor Lal.

**Data curation:** Dhruvendra Lal.

**Formal analysis:** Dhruvendra Lal.

**Methodology:** Dhruvendra Lal, Amrit Kaur Virk, Kavisha Kapoor Lal, Jayanta Bora.

**Project administration:** Dhruvendra Lal, Amrit Kaur Virk, Anu Bhardwaj, Kavisha Kapoor Lal, Sonu Goel.

**Resources:** Dhruvendra Lal, Sonu Goel.

**Software:** Dhruvendra Lal, Jayanta Bora.

**Validation:** Dhruvendra Lal, Amrit Kaur Virk, Anu Bhardwaj, Kavisha Kapoor Lal, Sonu Goel.

**Visualization:** Dhruvendra Lal, Jayanta Bora, Anuradha Nadda, Sonu Goel.

**Writing – original draft:** Dhruvendra Lal.

**Writing – review & editing:** Amrit Kaur Virk, Anu Bhardwaj, Kavisha Kapoor Lal, Jayanta Bora, Anuradha Nadda, Sonu Goel.

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
