## [Decision Letter · Decision Letter 0]

15 Jun 2023

PONE-D-23-07220Pestilent relationship between smoking and hypertension or pulse pressure among males over 15 years in India: NFHS-5 SurveyPLOS ONE

Dear Dr. Goel,

Thank you for submitting your manuscript to PLOS ONE. After careful consideration, we feel that it has merit but does not fully meet PLOS ONE’s publication criteria as it currently stands. Therefore, we invite you to submit a revised version of the manuscript that addresses the points raised during the review process.

We look forward to receiving your revised manuscript.

Kind regards,

Dorothy Lall

Academic Editor

PLOS ONE

Additional Editor Comments:

Thank you for the submission. The reviewer and myself feel there is merit in the manuscript but request a careful revision and response to the comments. Some additional comments

1. Please make the case stronger for why you are revisiting the association between smoking and blood pressure. The few studies cited 9,10,11 are not substantial - one was in a subset of post menopausal women, one is a 1986 paper...I would recommend instead to highlight the associations between smoker hypertensive and the social determinants variables that have been presented in the paper. This is much more useful to the current discourse and will be a contribution of this manuscript.

2. The discussion needs to include more on pulse pressure - what is the significance of pulse pressure, its biological plausibility etc.

3. State wise variation is mentioned- UP having highest prevalence of the smoker hypertensive , can be discussed

4. Methodology needs a section on analysis

5. Please be clear in the paper- you can only comment on associations not causality, so its not the effect of smoking on hypertension (discussion line 1) but only an association that is clear- the pathways are complex and we know they are not single variable dependant.

Reviewers' comments:

Reviewer's Responses to Questions

**Comments to the Author**

1. Is the manuscript technically sound, and do the data support the conclusions?

Reviewer #1: Partly

2. Has the statistical analysis been performed appropriately and rigorously? 

Reviewer #1: Yes

3. Have the authors made all data underlying the findings in their manuscript fully available?

Reviewer #1: Yes

4. Is the manuscript presented in an intelligible fashion and written in standard English?

Reviewer #1: Yes

5. Review Comments to the Author

Reviewer #1: Thank you for submitting this important paper.

I have some comments which can help you to improve the readability of the present paper.

1. Introduction: we need to make strong base why only males have been included in the present study, when we know smoking has more deleterious effects on females.

2. similarly why we have only considered smoking tobacco should be justified.

methods:

methods are vaguely written. we can consider using the format used by the corresponding author in the previous studies (https://journals.plos.org/plosone/article/file?id=10.1371/journal.pone.0233861&type=printable)

We need to reconsider as people taking medication for htn are not included in the sample selection process as per written methodology.

results: table 2 is very complex. we can have one top row to depict total counts. then, subsequently we can only depict weighted% to depict distribution of smokers as per HTN.

Also, we can have separate tables for HTN, and pulse pressure to give equal weightage to the two.

table 3: why separate odds have been calculated for narrow and wide PP, and if there is a clinical theory to support then it should be highlighted somewhere in early parts of the manuscript.

has the multicollinearity been checked for smoking daily, cigarette, beedi smoking variable (then what about other forms of smoking).

discussion: last line of 2nd para is very vague. it should be duly cited, or else it looks like a vague assumption.

not much has been mentioned about the independent variables chosen in the study in the methods section.

association and biological plausibility to depict association between pulse pressure and smoking has not been highlighted much in the discussion. like why the reader should be interested in pulse pressure when studying smoking. what are its consequence, how it is different from hypertension scientifically.

you can consider rewriting your discussion under the following sections:

Summary of main findings of your study (avoid repeating numbers and link to objectives)

What is new? Why important?

Strengths and limitations of the study

Compare to previous literature

Speculation where needed

Implications – programmatic and policy

Future research

6. PLOS authors have the option to publish the peer review history of their article (what does this mean?). If published, this will include your full peer review and any attached files.

Reviewer #1: **Yes: **Dr. Madhur Verma

---

## [Author Response · Author response to Decision Letter 0]

27 Jul 2023

27th July 2023

To 

The Editor 

Plos One Journal 

Respected sir/madam

I am writing this letter to express my heartfelt gratitude for your invaluable efforts and constructive feedback on our manuscript titled, “Pestilent relationship between smoking and hypertension or pulse pressure among males over 15 years in India: NFHS-5 Survey" submitted to Plos One. Your expertise and dedication have played a pivotal role in enhancing the quality and rigor of our work, and we are genuinely grateful for your time and commitment to this process.

Your insightful comments and thoughtful suggestions have been immensely valuable in refining our study and addressing its limitations. Your keen attention to detail and perceptive observations have challenged us to critically analyze our research, leading to significant improvements in the manuscript.

We deeply appreciate the thoroughness of your evaluation and the time you have taken to carefully review our work. Your expert guidance has been instrumental in shaping the manuscript into a more robust and scholarly contribution to the field. 

We have carefully considered each of the points raised in your evaluation and have addressed them comprehensively in the revised manuscript. Below, we present our responses to each of your comments:

S No Comments Responses

1. Please ensure that your manuscript meets PLOS ONE's style requirements, including those for file naming. The manuscript has been revised as per the guidelines issued by the journal. 

2. Please include a complete copy of PLOS’ questionnaire on inclusivity in global research in your revised manuscript. Our policy for research in this area aims to improve transparency in the reporting of research performed outside of researchers’ own country or community. A complete copy of PLOS’ questionnaire on inclusivity in global research has been included 

3. PLOS requires an ORCID iD for the corresponding author in Editorial Manager on papers submitted after December 6th, 2016. Please ensure that you have an ORCID iD and that it is validated in Editorial Manager. Included 

4. Your ethics statement should only appear in the Methods section of your manuscript. If your ethics statement is written in any section besides the Methods, please delete it from any other section. The ethics statement has been removed from all other places except for the Methods section.

5. We note that Figure 1 in your submission contain [map/satellite] images which may be copyrighted. All PLOS content is published under the Creative Commons Attribution License (CC BY 4.0), which means that the manuscript, images, and Supporting Information files will be freely available online, and any third party is permitted to access, download, copy, distribute, and use these materials in any way, even commercially, with proper attribution. For these reasons, we cannot publish previously copyrighted maps or satellite images created using proprietary data, such as Google software (Google Maps, Street View, and Earth). Fig 1 is a Bing Map under Microsoft which gives general rights for prints under Section 3 (Use with a Bing Maps Agreement), Section 4 (Use with a Volume Licensing Agreement) and Section 12 (Bing Maps API for Enterprise offerings purchased through the Azure Marketplace) of the TOU. 

The full print rights are available online at the following link: 

https://www.microsoft.com/en-us/maps/product/print-rights

 Additional Editor’s comments 

1. Please make the case stronger for why you are revisiting the association between smoking and blood pressure. The few studies cited 9,10,11 are not substantial - one was in a subset of post menopausal women, one is a 1986 paper...I would recommend instead to highlight the associations between smoker hypertensive and the social determinants variables that have been presented in the paper. This is much more useful to the current discourse and will be a contribution of this manuscript. Essential revisions have been diligently implemented, and we have also updated a few citations to incorporate articles that offer further substantiation of the association between smoking and hypertension.

2. The discussion needs to include more on pulse pressure - what is the significance of pulse pressure, its biological plausibility etc. The discussion section has undergone extensive and thorough revision, resulting in a comprehensive depiction and analysis of the pulse pressure findings and their implications. The revised manuscript now presents a detailed comparison of these findings with various studies available in the literature.

3. State wise variation is mentioned- UP having highest prevalence of the smoker hypertensive, can be discussed The prevalence of smoking and hypertension in Uttar Pradesh (UP) has been thoroughly examined and elaborated in the revised manuscript. Additionally, a comprehensive comparison has been drawn between the current findings and the data from NFHS-4 to provide valuable insights into the trends and changes in smoking and hypertension prevalence over time in this region.

4. Methodology needs a section on analysis Methodology section has been fully revised and analysis section has now been included. 

5. Please be clear in the paper- you can only comment on associations not causality, so its not the effect of smoking on hypertension (discussion line 1) but only an association that is clear- the pathways are complex and we know they are not single variable dependant. The same has been mentioned in the limitation part of the manuscript and first line of discussion portion has been edited accordingly. 

 Reviewer #1 Comments 

1. Introduction: we need to make strong base why only males have been included in the present study, when we know smoking has more deleterious effects on females. It has been discussed and included in the study participants section of Methods.

2. similarly why we have only considered smoking tobacco should be justified. The idea of including tobacco smoking for the present study has been described and included in the Introduction part of the revised manuscript. 

3. methods:

methods are vaguely written. we can consider using the format used by the corresponding author in the previous studies The Methods section of the manuscript has been meticulously revised, and subheadings have been thoughtfully incorporated to enhance clarity and organization. This thoughtful approach ensures that readers can easily navigate and comprehend the methodology employed in our study.

4. We need to reconsider as people taking medication for htn are not included in the sample selection process as per written methodology. The questionnaire is a part of NFHS-5 which includes 3 subsets. 

5. table 2 is very complex. we can have one top row to depict total counts. then, subsequently we can only depict weighted% to depict distribution of smokers as per HTN.

Also, we can have separate tables for HTN, and pulse pressure to give equal weightage to the two Table 2 has been thoughtfully redesigned to improve its comprehensibility and comparability. To reduce the number of tables and enhance the ease of comparison, the two subsets (hypertension and pulse pressure) have been amalgamated into a single table. This strategic consolidation allows for a more concise presentation of the data while ensuring that readers can readily assess and compare the various stages of hypertension and pulse pressure categories (Normal, Narrow, and Wide).

6. table 3: why separate odds have been calculated for narrow and wide PP, and if there is a clinical theory to support then it should be highlighted somewhere in early parts of the manuscript. The revised manuscript now comprehensively expounds on the distinct clinical implications of Narrow Pulse Pressure (PP) and Wide Pulse Pressure, recognizing their disparate significance in cardiovascular health. Owing to these divergent implications, separate odds ratios (OR) have been meticulously calculated for each category. This approach not only elucidates the unique associations of Narrow PP and Wide PP with various factors but also facilitates a more nuanced understanding of their respective impacts on cardiovascular outcomes.

7. discussion: last line of 2nd para is very vague. it should be duly cited, or else it looks like a vague assumption.

not much has been mentioned about the independent variables chosen in the study in the methods section. Discussion portion has been meticulously revised. 

8. association and biological plausibility to depict association between pulse pressure and smoking has not been highlighted much in the discussion. like why the reader should be interested in pulse pressure when studying smoking. what are its consequence, how it is different from hypertension scientifically. The difference and health implications of Pulse pressure has been described in the manuscript. 

9. you can consider rewriting your discussion Discussion part has been fully revised 

10 While revising your submission, please upload your figure files to the Preflight Analysis and Conversion Engine (PACE) digital diagnostic tool, https://pacev2.apexcovantage.com/. Editing using PACE has been done. 

In conclusion, we are grateful for the opportunity to address the issues raised by the academic editor and reviewer. We are confident that the revisions made in response to your invaluable feedback have significantly improved the manuscript's quality and scientific rigor.

Once again, we extend our sincere gratitude for your meticulous evaluation, which has undoubtedly strengthened the scholarly value of our work.

Thanking you 

Yours sincerely 

Authors

---

## [Editor Report · Decision Letter 1]

5 Oct 2023

PONE-D-23-07220R1Pestilent relationship between smoking and hypertension or pulse pressure among males over 15 years in India: NFHS-5 SurveyPLOS ONE

Dear Dr. Goel,

Thank you for submitting your manuscript to PLOS ONE. After careful consideration, we feel that it has merit but does not fully meet PLOS ONE’s publication criteria as it currently stands. Therefore, we invite you to submit a revised version of the manuscript that addresses the points raised during the review process.

We look forward to receiving your revised manuscript.

Kind regards,

Dorothy Lall

Academic Editor

PLOS ONE

Journal Requirements:

Additional Editor Comments:

1. Please give more details in the analysis. What was the regression model used, what factors were adjusted for and what was the strength of association after adjusting for age, obesity etc.

2. Consider rewriting the strengths, limitations and conclusion sections in scientific writing style..it currently reads like AI generated text!

---

## [Author Response · Author response to Decision Letter 1]

14 Oct 2023

Please review your reference list to ensure that it is complete and correct. If you have cited papers that have been retracted, please include the rationale for doing so in the manuscript text, or remove these references and replace them with relevant current references.  The references have been revisited and cited again in Vancouver styling and corrections have been made. There are no retracted articles. The reference list has been corrected. (Line 351 to 428)

Please give more details in the analysis. What was the regression model used, what factors were adjusted for and what was the strength of association after adjusting for age, obesity etc.  The data analysis part has been edited and more details have been added. (Line 144 to 158)

Consider rewriting the strengths, limitations, and conclusion sections in scientific writing style.  The strengths, limitations and conclusion sections have also been edited completely. (Line 322 to 344)

---

## [Editor Report · Decision Letter 2]

13 Nov 2023

Pestilent relationship between smoking and hypertension or pulse pressure among males over 15 years in India: NFHS-5 Survey

PONE-D-23-07220R2

Dear Dr. Goel,

We’re pleased to inform you that your manuscript has been judged scientifically suitable for publication and will be formally accepted for publication once it meets all outstanding technical requirements.

Kind regards,

Dorothy Lall

Academic Editor

PLOS ONE

Additional Editor Comments (optional):

Thank you all comments are addressed
---

## [Editor Report · Acceptance letter]

16 Nov 2023

PONE-D-23-07220R2 

Pestilent relationship between smoking and hypertension or pulse pressure among males over 15 years in India: NFHS-5 Survey. 

Dear Dr. Goel:

I'm pleased to inform you that your manuscript has been deemed suitable for publication in PLOS ONE. Congratulations! Your manuscript is now with our production department. 

Kind regards, 

on behalf of

Dr. Dorothy Lall 

Academic Editor

PLOS ONE